# Connections Between Mirror Descent, Thompson Sampling and the Information Ratio

**Julian Zimmert**
DeepMind, London/
University of Copenhagen
`zimmert@di.ku.dk`

**Tor Lattimore**
DeepMind, London
`lattimore@google.com`

## Abstract

The information-theoretic analysis by Russo and Van Roy [25] in combination with minimax duality has proved a powerful tool for the analysis of online learning algorithms in full and partial information settings. In most applications there is a tantalising similarity to the classical analysis based on mirror descent. We make a formal connection, showing that the information-theoretic bounds in most applications can be derived from existing techniques for online convex optimisation. Besides this, for $k$-armed adversarial bandits we provide an efficient algorithm with regret that matches the best information-theoretic upper bound and improve best known regret guarantees for online linear optimisation on $\ell_p$-balls and bandits with graph feedback.

## 1 Introduction

The combination of minimax duality and the information-theoretic machinery by Russo and Van Roy [25] has proved a powerful tool in the analysis of online learning algorithms. This has led to short and insightful analysis for $k$-armed bandits, linear bandits, convex bandits and partial monitoring, all improving on prior best known results. The downside is that the approach is non-constructive. The application of minimax duality demonstrates the existence of an algorithm with a given bound in the adversarial setting, but provides no way of constructing that algorithm.

The fundamental quantity in the information-theoretic analysis is the 'information ratio' in round $t$, which informally is

$$\text{information ratio}_t = \frac{(\text{expected regret in round } t)^2}{\text{expected information gain in round } t},$$

where the information gain is either measured using the mutual information [25] or a generalisation based on a Bregman divergence [21]. Proving the information ratio is small corresponds to showing that either the learner is suffering small regret in round $t$ or gaining information, which ultimately leads to a bound on the cumulative regret. The aforementioned generalisation by Lattimore and Szepesvári [21] (restated in the supplementary) lead to a short analysis for $k$-armed adversarial bandits that is minimax optimal except for small constant factors. The authors speculated that the new idea should lead to improved bounds for a range of online learning problems and suggested a number of applications, including bandits with graph feedback [3] and linear bandits on $\ell_p$-balls [11].

We started to follow this plan, successfully improving existing minimax bounds for bandits with graph feedback and online linear optimisation for $\ell_p$-balls with full information (the bandit setting remains a mystery). Along the way, however, we noticed a striking connection between the analysis techniques for bounding the information ratio and controlling the stability of online stochastic mirror descent (OSMD), which is a classical algorithm for online convex optimisation. A connection was

already hypothesised by Lattimore and Szepesvári [21], who noticed a similarity between the bounds obtained. Notably, why does using the negentropy potential in the information-theoretic analysis lead to almost identical bounds for $k$-armed bandits as Exp3? Why does this continue to hold with the Tsallis entropy and the INF strategy [6]?

**Contribution**   Our main contribution is a formal connection between the information-theoretic analysis and OSMD. Specifically, we show how tools for analysing OSMD can be applied to a modified version of Thompson sampling that uses the same sampling strategy as OSMD, but replaces the mirror descent update with a Bayesian update. This contribution is valuable for several reasons: (a) it explains the similarity between the information-theoretic and OSMD style analysis, (b) it allows for the transfer of techniques for OSMD to Bayesian regret analysis and (c) it opens the possibility of a constructive transfer of ideas from Bayesian regret analysis to the adversarial framework, as we illustrate in the next contribution.

A curiosity in the Bayesian analysis of adversarial $k$-armed bandits is that the resulting bound was always a factor of 2 smaller than the corresponding bound for OSMD. This was true in the original analysis [25] and its generalisation [21]. Our new theorem entirely explains the difference, and indeed, allows us to improve the bounds for OSMD. This leads to an efficient algorithm for adversarial $k$-armed bandits with regret $\mathfrak{R}_n \leq \sqrt{2kn} + O(k)$, matching the information-theoretic upper bound except for small lower-order terms.

Finally, we improve the regret guarantees for two online learning problems. First, for bandits with graph feedback we improve the minimax regret in the 'easy' setting by a $\log(n)$ factor, matching the lower bound up to a factor of $\log^{3/2}(k)$. Second, for online linear optimisation over the $\ell_p$-balls we improve existing bounds by arbitrarily large constant factors. At first we had proved these results using the information-theoretic tools and minimax duality, but here we present the unified view and consequentially the analysis also applies to OSMD for which we have efficient algorithms.

**Related work**   The information-theoretic Bayesian regret analysis was introduced by [24, 25, 26]. The focus in these papers is on the analysis of Bayesian algorithms in the stochastic setting, a line of work continued recently by [15]. [10] noticed that the stochastic assumption is not required and that the results continued to hold in a Bayesian adversarial setting where the prior is over arbitrary sequences of losses, rather than over (parametric) distributions as is usual in Bayesian statistics. The idea to use minimax duality to derive minimax regret bounds is due to [1] and has been applied and generalised by a number of authors [10, 17, 21, 9]. Mirror descent was developed by [22] and [23] for optimization. As far as we know its first application to bandits was by [2], which precipitated a flood of papers as summarised in the books by [8, 20]. We work in the partial monitoring framework, which goes back to [27]. Most of the focus since then has been on classifying the growth of the regret on the horizon for finite partial monitoring games [13, 16, 5, 7, 19]. Bandits with graph feedback are a special kind of partial monitoring problem and have been studied extensively [3, 14, 4, and others], with a monograph on the subject by [28]. Online linear optimisation is an enormous subject by itself. We refer the reader to the books by [12, 18].

**Notation**   The reader will find omitted proofs in the appendix. Let $[n] = \{1, 2, \ldots, n\}$ and $B_p^d = \{x \in \mathbb{R}^d : \|x\|_p \leq 1\}$ be the standard $\ell_p$-ball. For positive definite $A$ we write $\|x\|_A^2 = x^\top A x$. Given a topological space $X$, let $\text{int}(X)$ be its interior and $\Delta(X)$ be the space of probability measures on $X$ with the Borel $\sigma$-algebra. We write $X^\circ = \{y \in \mathbb{R}^d : \sup_{x \in X} |\langle x, y \rangle| \leq 1\}$ for the functional analysts polar and $\text{co}(X)$ for the convex hull of $X$. The domain of a convex function $F : \mathbb{R}^d \to \mathbb{R} \cup \{\infty\}$ is $\text{dom}(F) = \{x : F(x) < \infty\}$. For $x, y \in \text{dom}(F)$ the Bregman divergence between $x$ and $y$ with respect to $F$ is $D_F(x, y) = F(x) - F(y) - \nabla_{x-y} F(y)$ where $\nabla_v F(x)$ is the directional derivative of $F$ at $x$ in the direction $v$. The diameter of $X$ with respect to $F$ is $\text{diam}_F(X) = \sup_{x,y \in X} F(x) - F(y)$. We abuse notation by writing $\nabla^{-2} F(x) = (\nabla^2 F(x))^{-1}$. For $x, y \in \mathbb{R}^d$ we let $[x, y] = \text{co}(\{x, y\})$ be the convex hull of $x$ and $y$, which is the set of points on the chord between $x$ and $y$.

**Linear partial monitoring**   Our results are most easily expressed in a linear version of the partial monitoring framework, which extends the standard adversarial linear bandit framework to general feedback structures. Let $\mathcal{A}$ be the action space and $\mathcal{L}$ the loss space, which are subsets of $\mathbb{R}^d$ with $\mathcal{A}$ compact. The convex hull of $\mathcal{A}$ is $\mathcal{X} = \text{co}(\mathcal{A})$. When $\mathcal{A}$ is finite we let $k = |\mathcal{A}|$. The signal

function is a known function $\Phi : \mathcal{A} \times \mathcal{L} \to \Sigma$ for some observation space $\Sigma$. An adversary and learner interact over $n$ rounds. First the adversary secretly chooses $(\ell_t)_{t=1}^n$ with $\ell_t \in \mathcal{L}$ for all $t$. In each round $t$ the learner samples an action $A_t \in \mathcal{A}$ from a distribution depending on observations $A_1, \Phi_1, \ldots, A_{t-1}, \Phi_{t-1}$ where $\Phi_s = \Phi(A_s, \ell_s)$ is the observation in round $s$. The regret of policy $\pi$ in environment $(\ell_t)_{t=1}^n$ is

$$\mathfrak{R}_n(\pi, (\ell_t)_{t=1}^n) = \max_{a \in \mathcal{A}} \mathbb{E}\left[\sum_{t=1}^n \langle A_t - a, \ell_t \rangle\right],$$

where the expectation is with respect to the randomness in the actions. The regret depends on a policy and the losses. The minimax regret is

$$\mathfrak{R}_n^* = \inf_\pi \sup_{(\ell_t)_{t=1}^n} \mathfrak{R}_n(\pi, (\ell_t)_{t=1}^n),$$

where the infimum is over all policies and the supremum over all loss sequences in $\mathcal{L}^n$. From here on the dependence of $\mathfrak{R}_n$ on the policy and loss sequence is omitted.

**Examples** The standard $k$-armed bandit is recovered when $\mathcal{A} = \{e_1, \ldots, e_k\}$, $\mathcal{L} = [0, 1]^k$ and $\Phi(a, \ell) = \langle a, \ell \rangle \in \Sigma = [0, 1]$. For linear bandits the set $\mathcal{A}$ is an arbitrary compact set and $\mathcal{L}$ is typically $\mathcal{A}^\circ$. Bandits with graph feedback have a richer signal function as we explain in Section 4.

**Bayesian setting** In the Bayesian setting the sequence of losses $(\ell_t)_{t=1}^n$ are sampled from a known prior probability measure $\nu$ on $\mathcal{L}^n$ and subsequently the learner interacts with the sampled losses as normal. The optimal action is now a random variable $A^* = \arg\min_{a \in \mathcal{A}} \sum_{t=1}^n \langle a, \ell_t \rangle$ and the Bayesian regret is

$$\mathfrak{BR}_n = \mathbb{E}\left[\sum_{t=1}^n \langle A_t - A^*, \ell_t \rangle\right].$$

Finally, define $\mathbb{P}_t(\cdot) = \mathbb{P}(\cdot \mid \mathcal{F}_t)$ and $\mathbb{E}_t[\cdot] = \mathbb{E}[\cdot \mid \mathcal{F}_t]$ with $\mathcal{F}_t = \sigma(A_1, \Phi_1, \ldots, A_t, \Phi_t)$, $\Delta_t = \langle A_t - A^*, \ell_t \rangle$. A crucial piece of notation is $X_t = \mathbb{E}_{t-1}[A_t] \in \mathcal{X}$, which is the conditional expected action played in round $t$.

## 2 Mirror descent, Thompson sampling and the information ratio

We now develop the connection between OSMD and the information-theoretic Bayesian regret analysis. Specifically we show that instances of OSMD can be transformed into an algorithm similar to Thompson sampling (TS) for which the Bayesian regret can be bounded in the same way as the regret of the original algorithm. The similarity to TS is important. Any instance of OSMD with a uniform bound on the adversarial regret enjoys the same bound on the Bayesian regret for any prior without modification. Our

---

**Algorithm 1: OSMD**

**Input:** $\mathscr{A} = (P, E, F)$ and $\eta$
**Initialize** $X_1 = \arg\min_{a \in \mathcal{X}} F(a)$
**for** $t = 1, \ldots, n$ **do**
  Sample $A_t \sim P_{X_t}$ and observe $\Phi_t$
  Construct: $\hat{\ell}_t = E(X_t, A_t, \Phi_t)$
  Update: $X_{t+1} = f_t(X_t, A_t)$

---

result has a different flavour because we prove a bound for a variant of OSMD that replaces the mirror descent update with a Bayesian update.

OSMD is a modular algorithm that depends on defining three components: (1) A sampling scheme that determines how the algorithm explores, (2) a method for estimating the unobserved loss vectors, and (3) a convex 'potential' and learning rate that determines how the algorithm updates its iterates. The following definition makes this more precise.

**Definition 1.** An instance of OSMD is determined by a tuple $\mathscr{A} = (P, F, E)$ and learning rate $\eta > 0$ such that

(a) The sampling scheme is a collection $P = \{P_x : x \in \mathcal{X}\}$ of probability measures in $\Delta(\mathcal{A})$ such that $\mathbb{E}_{A \sim P_x}[A] = x$ for all $x \in \mathcal{X}$.

(b) The potential is a Legendre function $F : \mathbb{R}^d \to \mathbb{R} \cup \{\infty\}$ with $\mathrm{dom}(F) \cap \mathcal{X} \neq \emptyset$ and $\eta > 0$ is the learning rate.

(c) The estimation function is $E : \mathcal{X} \times \mathcal{A} \times \Sigma \to \mathbb{R}^d$, which we assume satisfies $\mathbb{E}_{A \sim P_x}[E(x, A, \Phi(A, \ell))] = \ell$ for all $\ell \in \mathcal{L}$ and $x \in \mathcal{X}$.

The assumptions on the mean of $P_x$ and that $E$ is unbiased are often relaxed in minor ways, but for simplicity we maintain the strict definition. For the remainder we fix $\mathscr{A} = (P, F, E)$ and $\eta > 0$ and abbreviate

$$E_t(x, a) = E(x, a, \Phi(a, \ell_t)) \qquad \text{and} \qquad \hat{\ell}_t = E(X_t, A_t, \Phi_t).$$

You should think of $E_t(x, a)$ as the estimated loss vector when the learner plays action $a$ while sampling from $P_x$ and $\hat{\ell}_t$ as the realisation of this estimate in round $t$. OSMD starts by initialising $X_1$ as the minimiser of $F$ constrained to $\mathcal{X}$. Subsequently it samples $A_t \sim P_{X_t}$ and updates

$$X_{t+1} = \arg\min_{y \in \mathcal{X}} \eta \langle y, \hat{\ell}_t \rangle + \mathrm{D}_F(y, X_t).$$

A useful notation is to let $(f_t)_{t=1}^n$ and $(g_t)_{t=1}^n$ be sequences of functions from $\mathcal{X} \times \mathcal{A}$ to $\mathbb{R}^d$ with

$$f_t(x, a) = \arg\min_{y \in \mathcal{X}} \left( \eta \langle y, E_t(x, a) \rangle + \mathrm{D}_F(y, x) \right) \qquad \text{and}$$

$$g_t(x, a) = \arg\min_{y \in \mathrm{int}(\mathrm{dom}(F))} \left( \eta \langle y, E_t(x, a) \rangle + \mathrm{D}_F(y, x) \right),$$

which means that $X_{t+1} = f_t(X_t, A_t)$, while $g_t$ is the same as $f_t$, but without the constraint to $\mathcal{X}$. The complete algorithm is summarised in Algorithm 1. The next theorem is well known [20, §28].

**Theorem 2** (OSMD REGRET BOUND). *The regret of OSMD satisfies*

$$\mathfrak{R}_n \leq \frac{\mathrm{diam}_F(\mathcal{X})}{\eta} + \frac{\eta}{2} \mathbb{E}\left[ \sum_{t=1}^n \mathrm{stab}_t(X_t; \eta) \right],$$

*where* $\mathrm{stab}_t(x; \eta) = \frac{2}{\eta} \mathbb{E}_{A \sim P_x}\left[ \langle x - f_t(x, A), E_t(x, A) \rangle - \frac{\mathrm{D}_F(f_t(x, A), x)}{\eta} \right].$

The random variable $\mathrm{stab}_t(X_t; \eta)$ measures the stability of the algorithm relative to the learning rate and is usually almost surely bounded. The diameter term depends on how fast the algorithm can move from the starting point to optimal, which is large when the learning rate is small. In this sense the learning rate is tuned to balance the stability of the algorithm and the requirement that $(X_t)$ can tend towards an optimal point. Note that $\mathrm{stab}_t(x)$ depends on $P$, $E$, $F$, $\eta$ and the loss vector $\ell_t$, which means that in the Bayesian setting the stability function is random. The next lemma is also known and is often useful for bounding the stability function.

**Lemma 3.** *Suppose that $F$ is twice differentiable on* $\mathrm{int}(\mathrm{dom}(F))$, *then*

$$\mathrm{stab}_t(x; \eta) \leq \mathbb{E}_{A \sim P_x}\left[ \sup_{z \in [x, f_t(x, A)]} \| E_t(x, A) \|^2_{\nabla^{-2}F(z)} \right].$$

*Furthermore, provided that $g_t(x, a)$ exists for all $a$ in the support of $P_x$, then*

$$\mathrm{stab}_t(x; \eta) \leq \mathbb{E}_{A \sim P_x}\left[ \sup_{z \in [x, g_t(x, A)]} \| E_t(x, A) \|^2_{\nabla^{-2}F(z)} \right].$$

**Bayesian analysis** Modified Thompson sampling (MTS) is a variant of TS summarised in Algorithm 2 that depends on a prior distribution $\nu$ and a sampling scheme $P$. The algorithm differs from Algorithm 1 in the computation of $X_t$. Rather than using the mirror descent update, it uses the Bayesian expected optimal action conditioned on the observations. Expectations in this subsection are with respect to both the prior and the actions, which means that

---

**Algorithm 2:** MTS

**Input:** Prior $\nu$ and $P$
**Initialize** $X_1 = \mathbb{E}[A^*]$
**for** $t = 1, \ldots, n$ **do**
    Sample $A_t \sim P_{X_t}$ and observe $\Phi_t$
    Update: $X_{t+1} = \mathbb{E}_{t-1}[A^*]$

---

$(\ell_t)_{t=1}^n$ are randomly distributed according to $\nu$ and consequently the functions $f_t$, $g_t$ and $\mathrm{stab}_t$ are random. Our main theorem is the following bound on the Bayesian regret of MTS.

**Theorem 4.** *MTS satisfies* $\mathfrak{BR}_n \leq \dfrac{\operatorname{diam}_F(\mathcal{X})}{\eta} + \dfrac{\eta}{2}\mathbb{E}\left[\sum_{t=1}^{n} \operatorname{stab}_t(X_t; \eta)\right].$

**Remark 5.** The stability function depends on $\mathscr{A} = (P, F, E)$ and $\eta$ while Algorithm 2 only uses $P$. In this sense Theorem 4 shows that MTS satisfies the given bound for all $E$, $F$ and $\eta$. MTS is the same as TS when sampling from the posterior is the same as sampling from $P_{X_t}$. A fundamental case where this always holds is when $\mathcal{A} = \{e_1, \ldots, e_d\}$ because each $x \in \mathcal{X}$ is uniquely represented as a linear combination of elements in $\mathcal{A}$ and hence $P_x$ is unique.

*Proof of Theorem 4.* Beginning with the definition of the per-step regret,

$$
\begin{aligned}
\mathbb{E}_{t-1}\left[\Delta_t\right] &= \langle X_t, \mathbb{E}_{t-1}[\ell_t]\rangle - \mathbb{E}_{t-1}\left[\langle A^*, \ell_t\rangle\right] \\
&= \langle X_t, \mathbb{E}_{t-1}[\hat{\ell}_t]\rangle - \mathbb{E}_{t-1}\left[\langle A^*, \hat{\ell}_t\rangle\right] &\text{(1)} \\
&= \langle X_t, \mathbb{E}_{t-1}[\hat{\ell}_t]\rangle - \mathbb{E}_{t-1}\left[\langle \mathbb{E}_{t-1}[A^* \mid A_t, \Phi_t], \hat{\ell}_t\rangle\right] &\text{(2)} \\
&= \mathbb{E}_{t-1}\left[\langle X_t - X_{t+1}, \hat{\ell}_t\rangle\right] &\text{(3)} \\
&\leq \mathbb{E}_{t-1}\left[\langle X_t - f_t(X_t, A_t), \hat{\ell}_t\rangle - \frac{1}{\eta}\operatorname{D}_F(f_t(X_t, A_t), X_t) + \frac{1}{\eta}\operatorname{D}_F(X_{t+1}, X_t)\right] &\text{(4)} \\
&\leq \mathbb{E}_{t-1}\left[\frac{\eta}{2}\operatorname{stab}_t(X_t; \eta) + \frac{1}{\eta}\operatorname{D}_F(X_{t+1}, X_t)\right]. &\text{(5)}
\end{aligned}
$$

Eq. (1) uses that the loss estimators are unbiased. Eq. (2) follows using the tower rule for conditional expectations and the fact that $\hat{\ell}_t$ is a measurable function of $X_t$, $A_t$ and $\Phi_t$ so that

$$
\mathbb{E}_{t-1}[\langle A^*, \hat{\ell}_t\rangle] = \mathbb{E}_{t-1}[\mathbb{E}_{t-1}[\langle A^*, \hat{\ell}_t\rangle \mid A_t, \Phi_t]] = \mathbb{E}_{t-1}[\langle \mathbb{E}_{t-1}[A^* \mid A_t, \Phi_t], \hat{\ell}_t\rangle] = \mathbb{E}_{t-1}[\langle X_{t+1}, \hat{\ell}_t\rangle].
$$

Eq. (3) uses the definitions of $X_{t+1}$. Eq. (4) follows from the definition of $f_t$, which implies that

$$
\langle f_t(X_t, A_t), \hat{\ell}_t\rangle + \frac{1}{\eta}\operatorname{D}_F(f_t(X_t, A_t), X_t) \leq \langle X_{t+1}, \hat{\ell}_t\rangle + \frac{1}{\eta}\operatorname{D}_F(X_{t+1}, X_t).
$$

Finally, Eq. (5) follows from the definition of $\operatorname{stab}_t$. The proof is completed by summing over the per-step regret, noting that $(X_t)_{t=1}^n$ is a $(\mathcal{F}_t)_t$-adapted martingale and by [21, Theorem 3],

$$
\mathbb{E}\left[\sum_{t=1}^{n}\operatorname{D}_F(X_{t+1}, X_t)\right] \leq \mathbb{E}[F(X_{n+1})] - F(X_1) \leq \operatorname{diam}_F(\mathcal{X}). \qquad \square
$$

**The stability coefficient** The only difference between Theorems 2 and 4 is the trajectory of $(X_t)_{t=1}^n$ and the randomness of the stability function. In most analyses of OSMD the final bound is obtained via a uniform bound on $\operatorname{stab}_t(x; \eta)$ that holds regardless of the losses and in this case the trajectory $X_t$ is irrelevant. This is formalised in the following definition and corollary. Define the stability coefficients by

$$
\operatorname{stab}(\mathscr{A}; \eta) = \sup_{x \in \mathcal{X}} \max_{t \in [n]} \operatorname{stab}_t(x; \eta) \qquad \text{and} \qquad \operatorname{stab}(\mathscr{A}) = \sup_{\eta > 0} \operatorname{stab}(\mathscr{A}; \eta).
$$

**Corollary 6.** *The regret of Algorithm 1 for an appropriately tuned learning rate is bounded by*

$$
\mathfrak{R}_n \leq \sqrt{2\operatorname{diam}_F(\mathcal{X})\operatorname{stab}(\mathscr{A})n}.
$$

*The Bayesian regret of Algorithm 2 is bounded by* $\mathfrak{BR}_n \leq \sqrt{2\operatorname{diam}_F(\mathcal{X})\operatorname{ess\,sup}(\operatorname{stab}(\mathscr{A}))n}.$

The essential supremum is needed because the stability coefficient depends on the losses $(\ell_t)_{t=1}^n$, which are random in the Bayesian setting. Generally speaking, however, bounds on the stability coefficient are proven in a manner that is independent of the losses.

**Remark 7.** Often $\operatorname{stab}(\mathscr{A}; \eta) \leq a + b\eta$ for constants $a, b \geq 0$ and $\operatorname{stab}(\mathscr{A}) = \infty$. Nevertheless, the same argument shows that the regret of Algorithm 1 is bounded by

$$
\mathfrak{R}_n \leq \sqrt{2a\operatorname{diam}_F(\mathcal{X})n} + \frac{b\operatorname{diam}_F(\mathcal{X})}{a},
$$

and similarly for the Bayesian regret of Algorithm 2.

**Stability and the information ratio** The generalised information-theoretic analysis by [21] starts by assuming there exists a constant $\alpha > 0$ such that the following bound on the information ratio holds almost surely:

$$\text{information ratio}_t = \mathbb{E}_{t-1}[\Delta_t]^2 \Big/ \mathbb{E}_{t-1}[\mathrm{D}_F(X_{t+1}, X_t)] \leq \alpha . \tag{6}$$

Then [21, Theorem 3] shows that

$$\mathfrak{BR}_n \leq \sqrt{\alpha n \operatorname{diam}_F(\mathcal{X})} . \tag{7}$$

The proof of Theorem 4 directly provides a bound on the information ratio in terms of the stability coefficient. To see this, notice that Eq. (5) holds for all measurable $\eta$ and let

$$\eta = \sqrt{2\mathbb{E}_{t-1}[\mathrm{D}_F(X_{t+1}, X_t)]/\operatorname{ess\,sup}(\operatorname{stab}(\mathscr{A}))} . \tag{8}$$

Then by Eq. (5) and the definition of $\operatorname{stab}(\mathscr{A})$ it follows that

$$\mathbb{E}_{t-1}[\Delta_t]^2 \Big/ \mathbb{E}_{t-1}[\mathrm{D}_F(X_{t+1}, X_t)] \leq 2\operatorname{ess\,sup}(\operatorname{stab}(\mathscr{A})) \ \ a.s..$$

In other words, the usual methods for bounding the stability coefficient in the analysis of OSMD can be used to bound the information ratio in the information-theoretic analysis.

**Example 8.** To make the abstraction more concrete, consider the $k$-armed bandit problem where $\mathcal{L} = [0, 1]^k$ and $\mathcal{A} = \{e_1, \ldots, e_k\}$. In this case there is a unique sampling scheme defined by $P_x(a) = \langle x, a \rangle$. The standard loss estimation function is to use importance-weighting, which leads to

$$E_t(x, a)_i = \ell_{ti} \mathbb{1}(a = e_i)/x_i . \tag{9}$$

A commonly used potential is the unnormalised negentropy $F(x) = \sum_{i=1}^k x_i \log(x_i) - x_i$ that satisfies $\nabla^{-2} F(x) = \operatorname{diag}(x)$. The instance of OSMD resulting from these choices is called Exp3 for which an explicit form for $X_t$ is well known:

$$X_{ti} = \exp\left(-\eta \sum_{s=1}^{t-1} \hat{\ell}_{si}\right) \Big/ \left(\sum_{j=1}^k \exp\left(-\eta \sum_{s=1}^{t-1} \hat{\ell}_{sj}\right)\right) .$$

A short calculation shows that $g_t(x, a)_i = x_i \exp(-\eta \hat{\ell}_{ti}) \leq x_i$. The stability function is bounded using the second part of Lemma 3 by

$$\operatorname{stab}_t(x; \eta) \leq \mathbb{E}_{A \sim P_x}\left[\sup_{z \in [x, g_t(x,A)]} \|E_t(x, A)\|_{\nabla^{-2} F(z)}^2\right]$$

$$= \mathbb{E}_{A \sim P_x}\left[\sup_{z \in [x, g_t(x,A)]} \sum_{i=1}^k z_{ti} \frac{\mathbb{1}(A = e_i)\ell_{ti}^2}{x_{ti}^2}\right] = \mathbb{E}_{A \sim P_x}\left[\frac{\mathbb{1}(A = e_i)\ell_{ti}^2}{x_{ti}}\right] \leq \sum_{i=1}^k \ell_{ti}^2 \leq k .$$

Finally, the diameter of the probability simplex $\mathcal{X}$ with respect to the unnormalised negentropy is $\operatorname{diam}_F(\mathcal{X}) = \log(k)$. Applying Theorem 2 shows that the regret of OSMD and Bayesian regret of MTS satisfy

$$\mathfrak{R}_n \leq \sqrt{2nk \log(k)} \quad \text{(OSMD)} \qquad \text{and} \qquad \mathfrak{BR}_n \leq \sqrt{2nk \log(k)} \quad \text{(MTS)} .$$

**Remark 9.** Theorems 2 and 4 are vacuous when $\operatorname{diam}_F(\mathcal{X}) = \infty$. The most straightforward resolution is to restrict $X_t$ to a subset of $\mathcal{X}$ on which the diameter is bounded and then control the additive error. This idea also works in the Bayesian setting as described by [21]. We omit a detailed discussion to avoid technicalities.

## 3 Bandits

The best known bound on the minimax regret for $k$-armed bandits is $\mathfrak{R}_n \leq \sqrt{2kn}$ by [21]. They let $F(x) = -2\sum_{i=1}^k \sqrt{x_i}$ be the $1/2$-Tsallis entropy and prove that

$$\mathbb{E}_{t-1}[\Delta_t]^2 \Big/ \mathbb{E}_{t-1}[\mathrm{D}_F(X_{t+1}, X_t)] \leq \sqrt{k} .$$

By Cauchy-Schwarz $\operatorname{diam}_F(\mathcal{X}) \leq 2\sqrt{k}$ and then Eq. (7) shows that $\mathfrak{BR}_n \leq \sqrt{2nk}$ for all priors $\nu$. Minimax duality is used to conclude that $\mathfrak{R}_n^* \leq \sqrt{2kn}$. Meanwhile, using the importance-weighted estimator in Eq. (9) leads to a bound on the stability coefficient of $\operatorname{stab}(\mathscr{A}) \leq 2\sqrt{k}$ and then Theorem 2 yields a bound of $\mathfrak{R}_n \leq \sqrt{8nk}$.

The discrepancy between these methods is entirely explained by the naive choice of importance-weighted estimator. The approach based on bounding the information ratio is effectively shifting the losses, which can be achieved in the OSMD framework by shifting the importance-weighted estimators (see Fig. 1). This idea reduces the worst-case variance of the importance weighted estimators by a factor of 4.

**Lemma 10.** *If the loss estimator in Example 8 with* $F(s) = -2\sum_{i=1}^{k}\sqrt{x_i}$ *is replaced by*

$$E_t(x,a)_i = \frac{(\ell_{ti} - c_{ti})\mathbb{1}(a = e_i)}{x_i} + c_{ti},$$

*where* $c_{ti} = \frac{1}{2}(1 - \mathbb{1}(X_{ti} < \eta^2))$,

*then the stability coefficient for any* $\eta \leq 1/2$ *is bounded by* $\mathrm{stab}(\mathscr{A}; \eta) \leq k^{1/2}/2 + 12k\eta$.

**Theorem 11.** *The regret of OSMD with the loss estimator of Lemma 10 and appropriate learning rate satisfies:* $\mathfrak{R}_n \leq \sqrt{2kn} + 48k$.

## 4  Bandits with graph feedback

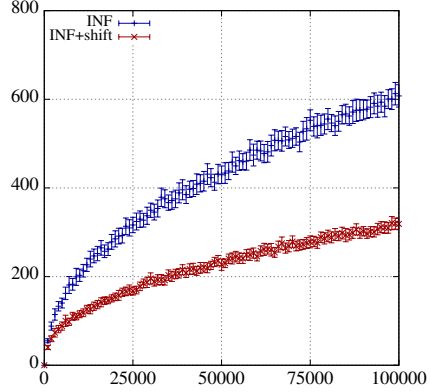

Figure 1: Comparison of INF with and without shifted loss estimators. $x$-axis is number of time-steps and $y$-axis the empirical regret estimation. $\eta$ is tuned to the horizon and all experiments use Bernoulli losses with $\mathbb{E}[\ell_t] = (0.45, 0.55, \ldots, 0.55)^T$ ($k = 5$). We repeat the experiment 100 times with error bars indicating three standard deviations. The empirical result matches our theoretical improvement of a factor 2.

In bandits with graph feedback the action set is $\mathcal{A} = \{e_1, \ldots, e_k\}$ and $\mathcal{L} = [0,1]^k$. Let $E \subseteq [k] \times [k]$ be a set of directed edges over vertex set $[k]$ so that $\mathcal{G} = ([k], E)$ is a directed graph. The signal function is $\Phi(e_i, \ell) = \{(j, \ell_j) : j \in \mathcal{N}(i)\}$. The standard bandit framework is recovered when $E = \{(i,i) : i \in [k]\}$ while the full information setup corresponds to $E = [k] \times [k]$. Of course there are settings between and beyond these extremes. The difficulty of the graph feedback problem is determined by the connectivity of the graph. For example, when $E = \emptyset$, the learner has no way to estimate the losses and the regret is linear in the worst case. Like finite partial monitoring, graph feedback problems can be classified into one of four regimes for which:

$$\mathfrak{R}_n^* \in \left\{ \mathcal{O}(1), \ \tilde{\Theta}(n^{1/2}), \ \Theta(n^{2/3}), \ \Omega(n) \right\}.$$

Our focus is on graph feedback problems that fit in the second category, which is the most challenging to analyse.

**Definition 12.** $\mathcal{G}$ *is called strongly observable if for every vertex* $i \in [k]$ *at least one of the following holds: (a)* $a \in \mathcal{N}(b)$ *for all* $b \neq a$ *or (b)* $a \in \mathcal{N}(a)$.

Alon et al. [3] prove the minimax regret for bandits with graph feedback is $\tilde{\Theta}(n^{1/2})$ if and only if $k > 1$ and $\mathcal{G}$ is strongly observable. They also prove the following theorem upper and lower bounding the dependence of the minimax regret on the horizon, the number of actions and a graph functional called the independence number.

**Theorem 13** ([3]). *Let* $\mathcal{G}_{ind}$ *be the independence number of* $\mathcal{G}$, *which is the cardinality of the largest subset of vertices such that no tow distinct vertices are connected by an edge. Suppose* $k > 1$ *and* $\mathcal{G}$ *is strongly observable. Then* $\mathfrak{R}_n^* = \mathcal{O}(\sqrt{\mathcal{G}_{ind}n}\log(kn))$ *and* $\mathfrak{R}_n^* = \Omega(\sqrt{\mathcal{G}_{ind}n})$.

The logarithmic dependence on $n$ in the proof of Theorem 13 appears quite naturally, which raises the question of whether or not the upper or lower bound is tight. In fact, as $n$ tends to infinity the upper bound in Theorem 13 could be improved to $\mathcal{O}(\sqrt{nk})$ by using a finite-armed algorithm that ignores the feedback except for the played action. Perhaps the independence number is not as fundamental as first thought? The following theorem shows the upper bound can be improved.

**Theorem 14.** *Let* $\mathscr{A} = (P, E, F)$ *be a triple defining OSMD with* $P_x(a) = \langle a, x \rangle$,

$$F(x) = \frac{1}{\alpha(1-\alpha)} \sum_{i=1}^{k} x_i^{\alpha} \qquad \text{where} \quad \alpha = 1 - 1/\log(k).$$

*Finally, define the unbiased loss estimation function E by*

$$E_t(x,a)_i = \frac{\ell_{ti}\mathbb{1}(a \in \mathcal{N}(i))}{\sum_{b \in \mathcal{N}(i)} x_b} \text{ for } i \notin I_t, \text{ and } E_t(x,a)_i = \frac{(\ell_{ti}-1)\mathbb{1}(a \neq i)}{1-x_i} + 1 \text{ otherwise },$$

*where $I_t = \{i \in [k] : i \notin \mathcal{N}(i) \text{ and } X_{ti} > 1/2\}$. Then for any $k \geq 8$ and an appropriately tuned learning rate the regret of OSMD with $\mathscr{A}$ satisfies $\mathfrak{R}_n = \mathcal{O}(\sqrt{\mathcal{G}_{ind} n \log(k)^3})$.*

## 5 Online linear optimisation over $\ell_p$-balls

We now consider full information online linear optimization on the $\ell_p$ balls with $p \in [1,2]$, which is modelled in our framework by choosing $\mathcal{A} = B_p^d$ and $\mathcal{L} = B_q^d$ with $1/p + 1/q = 1$ and $\Phi(a,\ell) = \ell$. Table 1 summarises the known results. When $p = 1$ the situation is unambiguous, with matching upper and lower bounds. For $p \in (1,2]$ there exist algorithms for which the regret is dimension free, but with constants that become arbitrarily large as $p$ tends to 1. Known results for online gradient descent (OGD) prove the blowup in terms of $p$ is avoidable, but with a price that is polynomial in the dimension.

| $p$ | Regret | Algorithm |
|---|---|---|
| $p = 1$ | $\sqrt{n\log(d)}$ | Hedge |
| $p > 1$ | $\sqrt{n/(p-1)}$ | [12, §11.5] |
| $p \geq 1$ | $\sqrt{d^{2/p-1}n}$ | OGD [18] |

Table 1: Known results for $\ell_p$-balls

**Theorem 15.** *For any $p \in [1,2]$, let h be the following convex and twice continuously differentiable function:*

$$h(x) = \begin{cases} \frac{d}{2}x^2 & \text{if } |x| \leq d^{\frac{1}{p-2}} \\ \frac{p-2}{p-1}d^{\frac{p-1}{p-2}}|x| + \frac{|x|^p}{p(p-1)} + \frac{2-p}{2p}d^{\frac{p}{p-2}} & \text{otherwise}. \end{cases}$$

*Then for OSMD using potential $F(x) = \sum_{i=1}^d h(x_i)$, loss estimator $E(x,a,\sigma) = \sigma$, an arbitrary exploration scheme and appropriately tuned learning rate,*

$$\mathfrak{R}_n = \mathcal{O}\left(\sqrt{\min\{1/(p-1),\log(d)\}\,n}\right).$$

*Furthermore, the Bayesian regret of TS is bounded by the same quantity.*

**Remark 16.** In the full information setting the loss estimation is independent of the action, which explains the arbitrariness of the exploration scheme. The intuitive justification for the slightly cryptic potential function is provided in the supplementary material.

## 6 Discussion

We demonstrated a connection between the information-theoretic analysis and OSMD. For $k$-armed bandits, we explained the factor of two difference between the regret analysis using information-theoretic and convex-analytic machinery and improved the bound for the latter. For graph bandits we improved the regret by a factor of $\log(n)$. Finally, we designed a new potential for which the regret for online linear optimisation over the $\ell_p$-balls improves the previously best known bound by arbitrarily large constant factors.

**Open problems** The main open problem is whether or not we can 'close the circle' and use the information-theoretic analysis to directly construct OSMD algorithms. Another direction is to try and relax the assumption that the loss is linear. The leading constant in the new bandit analysis now matches the best known information-theoretic bound [21]. There is still a constant lower-order term, which presently seems challenging to eliminate. In bandits with graph feedback one can ask whether the $\log(k)$ dependency can be improved. Lower bounds are still needed for $\ell_p$-balls and extending the idea to the bandit setting is an obvious followup. Finally, the best known algorithms for finite partial monitoring also use the information-theoretic machinery. Understanding how to borrow the ideas for OSMD remains a challenge.

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
