[Supplementary Material]

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

## A  Theorem 3 of [21]

**Theorem.** Let $(M_t)_{t=1}^{n+1}$ be an $\mathbb{R}^d$-valued martingale adapted to $(\mathcal{F}_t)_{t=1}^{n+1}$ and $M_t \in \mathcal{X} \subset \mathbb{R}^d$ almost surely for all $t$. Then let $F$ be a convex function with $\operatorname{diam}_F(\mathcal{X}) < \infty$. Suppose there exist constants $\alpha, \beta \geq 0$ such that $\mathbb{E}_t[\Delta_t] \leq \alpha + \sqrt{\beta \mathbb{E}_t[D_F(M_{t+1}, M_t)]}$ almost surely for all $t$. Then $\mathfrak{BR}_n \leq \alpha n + \sqrt{n\beta \operatorname{diam}_F(\mathcal{X})}$.

## B  Proof of Lemma 3

The proof is rather standard. In fact, the first part is [20, Theorem 26.13]. For the second part, fix $x \in \mathcal{X}$ and $a \in \mathcal{A}$ and define

$$\Psi(y) = \eta \langle y, E_t(x,a) \rangle + D_F(y,x) \,.$$

By the assumption that $g_t(x,a) \in \operatorname{int}(\operatorname{dom}(F)) = \operatorname{int}(\operatorname{dom}(\Psi))$ and the definition of $g_t(x,a)$ as the minimizer of $\Psi$ it follows that

$$0 = \nabla\Psi(g_t(x,a)) = \eta E_t(x,a) + \nabla F(g_t(x,a)) - \nabla F(x) \,.$$

Hence

$$
\begin{aligned}
\operatorname{stab}_t(x) &= \frac{2}{\eta}\mathbb{E}_{A \sim P_x}\left[\langle x - f_t(x,A), E_t(x,A) \rangle - \frac{D_F(f_t(x,A),x)}{\eta}\right] \\
&= \frac{2}{\eta}\mathbb{E}_{A \sim P_x}\left[\frac{1}{\eta}\langle x - f_t(x,A), \nabla F(x) - \nabla F(g_t(x,a)) \rangle - \frac{D_F(f_t(x,A),x)}{\eta}\right] \\
&= \frac{2}{\eta}\mathbb{E}_{A \sim P_x}\left[\frac{1}{\eta}D_F(x, g_t(x,A)) - \frac{1}{\eta}D_F(f_t(x,a), g_t(x,A))\right] \\
&\leq \frac{2}{\eta}\mathbb{E}_{A \sim P_x}\left[\frac{D_F(x, g_t(x,A))}{\eta}\right] \,.
\end{aligned}
\tag{10}
$$

Let $F^*$ be the Legendre dual of $F$. Since $F$ is Legendre and twice differentiable on $\operatorname{int}(\operatorname{dom}(F))$ it follows from Taylor's theorem and duality that there exists a $z^* \in [\nabla F(x), \nabla F(x) - \eta E_t(x,a)]$

such that

$$
\begin{aligned}
\mathrm{D}_F(x, g_t(x, a)) &= \mathrm{D}_{F^*}(\nabla F(g_t(x, a)), \nabla F(x)) \\
&= \mathrm{D}_{F^*}(\nabla F(x) - \eta E_t(x, a), \nabla F(x)) \\
&= \frac{\eta^2}{2} \|E_t(x, a)\|^2_{\nabla^2 F^*(z^*)} \\
&= \frac{\eta^2}{2} \|E_t(x, a)\|^2_{\nabla^{-2} F(\nabla F^*(z^*))} \\
&\leq \sup_{z \in [x, g_t(x, a)]} \frac{\eta^2}{2} \|E_t(x, a)\|^2_{\nabla^{-2} F(z)}.
\end{aligned}
$$

Substituting into Eq. (10) completes the result.

**Refined bound for the probability simplex**  For the proofs in the next sections, we require a refined version of Lemma 3. Let $1_k$ denote the vector with all ones.

**Lemma 17.** *Assume that* $\mathcal{A} = \{e_1, \ldots, e_k\}$ *and for* $c \in \mathbb{R}$ *define*

$$
f_{tc}(x, a) = \arg\min_{y \in \mathcal{X}} \left( \eta \langle y, E_t(x, a) + c1_k \rangle + \mathrm{D}_F(y, x) \right),
$$

$$
g_{tc}(x, a) = \arg\min_{y \in \mathrm{int}(\mathrm{dom}(F))} \left( \eta \langle y, E_t(x, a) + c1_k \rangle + \mathrm{D}_F(y, x) \right).
$$

*Provided that* $g_{tc}(x, a)$ *exists for all* $a$ *in the support of* $P_x$,

$$
\mathrm{stab}_t(x; \eta) \leq \frac{2}{\eta^2} \mathbb{E}_{A \sim P_x} \left[ \mathrm{D}_F(x, g_{tc}(x, A)) \right] \leq \mathbb{E}_{A \sim P_x} \left[ \sup_{z \in [x, g_{tc}(x, A)]} \|E_t(x, A) + c1_k\|^2_{\nabla^{-2} F(z)} \right].
$$

*Proof.* Since $\mathcal{X}$ is the probability simplex $\langle y, c1_k \rangle = c$ for all $y \in \mathcal{X}$. Therefore $f_{tc}(x, a) = f_t(x, a)$ and $\langle x - f_t(x, a), c1_k \rangle = 0$. Hence

$$
\begin{aligned}
\mathrm{stab}_t(x) &= \frac{2}{\eta} \mathbb{E}_{A \sim P_x} \left[ \langle x - f_t(x, A), E_t(x, A) \rangle - \frac{\mathrm{D}_F(f_t(x, A), x)}{\eta} \right] \\
&= \frac{2}{\eta} \mathbb{E}_{A \sim P_x} \left[ \langle x - f_{tc}(x, A), E_t(x, A) + c1_k \rangle - \frac{\mathrm{D}_F(f_{tc}(x, A), x)}{\eta} \right].
\end{aligned}
$$

The remaining proof is analogous to the proof of Lemma 3 substituting $f_t, g_t$ by $f_{tc}, g_{tc}$ and the loss $E_t(x, a)$ by $E_t(x, a) + c1_k$. $\qquad\square$

## C  Proof of Corollary 6

Starting with the adversarial regret bound. By Theorem 2,

$$
\mathfrak{R}_n \leq \frac{\mathrm{diam}_F(\mathcal{X})}{\eta} + \frac{\eta}{2} \mathbb{E} \left[ \sum_{t=1}^n \mathrm{stab}_t(X_t) \right] \leq \frac{\mathrm{diam}_F(\mathcal{X})}{\eta} + \frac{\eta n \, \mathrm{stab}(\mathscr{A})}{2}.
$$

The first part follows by choosing

$$
\eta = \sqrt{\frac{2 \, \mathrm{diam}_F(\mathcal{X})}{n \, \mathrm{stab}(\mathscr{A})}}.
$$

The Bayesian case follows from an identical argument and Theorem 4 and the fact that

$$
\mathbb{E} \left[ \sum_{t=1}^n \mathrm{stab}_t(X_t) \right] \leq \mathbb{E} \left[ \sum_{t=1}^n \mathrm{stab}(\mathscr{A}) \right] \leq n \, \mathrm{ess\,sup}(\mathrm{stab}(\mathscr{A})).
$$

The result claimed in Remark 7 follows similarly with the same choice of learning rate.

# D Proof of Theorem 11

*Proof of Lemma 10.* We use Lemma 17 with $c = -\frac{1}{2}$. As a reminder, we have

$$E_t(x, a)_i + c = \frac{(\ell_{ti} - c_{ti})\mathbb{1}(a = e_i)}{x_i} + c_{ti} + c, \text{ where } c_{ti} = \frac{1}{2}(1 - \mathbb{1}(X_{ti} < \eta^2)).$$

Let $\tilde{\ell}_t = E_t(X_t, A_t) + c\mathbb{1}_k$. We start by calculating the Hessian of $F$. Since $F(a) = -\sum_{i=1}^k 2\sqrt{a_i}$,

$$\nabla F(a) = -1/\sqrt{a} \qquad \text{and} \qquad \nabla^2 F(a) = \mathrm{diag}(a^{-3/2}/2).$$

The next step is to bound $g_{tc}(X_t, A_t)_i^{\frac{3}{2}}$. By definition

$$g_{tc}(X_t, A_t) = \underset{y \in \mathrm{int}(\mathrm{dom}(F))}{\arg\min} \eta\langle y, \tilde{\ell}_t\rangle + F(y) - F(X_t) - \langle y - X_t, \nabla F(X_t)\rangle,$$

which implies that $\eta\tilde{\ell}_t + \nabla F(g_{tc}(X_t, A_t)) - \nabla F(X_t) = 0$. Substituting the gradient of the potential shows that

$$\eta\tilde{\ell}_{ti} - \frac{1}{\sqrt{g_{tc}(X_t, A_t)_i}} + \frac{1}{\sqrt{X_{ti}}} = 0.$$

Solving for $g_{tc}(X_t, A_t)_i$ yields

$$g_{tc}(X_t, A_t)_i^{\frac{3}{2}} = \frac{X_{ti}^{\frac{3}{2}}}{(1 + \tilde{\ell}_t \eta X_{ti}^{\frac{1}{2}})^3}. \tag{11}$$

For $\tilde{\ell}_{ti} \geq 0$, Eq. (11) directly implies $g_{tc}(X_t, A_t)_i^{\frac{3}{2}} \leq X_{ti}^{\frac{3}{2}}$. Let $\tilde{\ell}_{ti} < 0$, then we get the following lower bound by definition of $\tilde{\ell}_t$:

$$X_{ti} \geq \eta^2 : \tilde{\ell}_{ti} = -\frac{(\ell_{ti} - 1)\mathbb{1}(A_t = e_i)}{2X_{ti}} \geq -\frac{1}{2X_{ti}} \geq -\frac{1}{2\eta X_{ti}^{1/2}},$$

$$X_{ti} < \eta^2 : \tilde{\ell}_{ti} = \frac{\ell_{ti}\mathbb{1}(A_t = e_i)}{X_{ti}} - \frac{1}{2} \geq -\frac{1}{2\eta X_{ti}^{1/2}} \geq -\frac{1}{2X_{ti}}.$$

This directly implies $-\tilde{\ell}_{ti}\eta X_{ti}^{1/2} \leq \frac{1}{2}\eta X_{ti}^{-1/2}$ and $1 + \tilde{\eta}X_{ti}^{1/2} \geq \frac{1}{2}$. Going back to Eq. (11), the following bound on $f(x) = x^{-3}$ holds due to convexity for all $x > -1$: $f(1+x) \leq f(1)+xf'(1+x)$. Using all three inequalities provides the bound

$$X_{ti}^{\frac{3}{2}}(1 + \tilde{\ell}_{ti}\eta X_{ti}^{\frac{1}{2}})^{-3} \leq X_{ti}^{\frac{3}{2}}\left(1 - 3(1 + \tilde{\ell}_{ti}\eta X_{ti}^{\frac{1}{2}})^{-4}\tilde{\ell}_{ti}\eta X_{ti}^{\frac{1}{2}}\right) \leq X_{ti}^{\frac{3}{2}} + 24\eta X_{ti}.$$

Hence for any $z \in [X_t, g_{tc}(X_t, A_t)]$ we have

$$\nabla^{-2}F(z) \preceq \mathrm{diag}(2X_t^{\frac{3}{2}} + 48\eta X_t \circ \mathbb{1}(\tilde{\ell}_t < 0)),$$

where $\mathbb{1}(\tilde{\ell}_t > 0)$ is vector of element wise applied indicator function. Finally we are ready to bound the stability:

$$\mathbb{E}_{A \sim P_{X_t}}\left[\sup_{z \in [X_t, g_{tc}(X_t, A)]} \|E_t(X_t, A) + c\mathbb{1}_k\|^2_{\nabla^{-2}F(z)}\right]$$

$$\leq \sum_{i:X_{ti}\geq\eta^2} X_{ti}\frac{(\ell_{ti} - \frac{1}{2})^2}{X_{ti}^2}(2X_{ti}^{\frac{3}{2}} + 48\eta X_{ti}) + \sum_{i:X_{ti}<\eta^2} \frac{1}{2^2}(2X_{ti}^{\frac{3}{2}} + 48\eta X_{ti}) + X_{ti}\frac{\ell_{ti}^2}{X_{ti}^2}2X_{ti}^{\frac{3}{2}} \tag{12}$$

$$\leq \sum_{i:X_{ti}\geq\eta^2} \frac{X_{ti}^{\frac{1}{2}}}{2} + 12\eta + \sum_{i:X_{ti}<\eta^2} \frac{25\eta^3}{2} + 2\eta \leq \frac{\sqrt{k}}{2} + 12\eta k. \tag{13}$$

Eq. (12) follows because for $X_{ti} \geq \eta^2$ the term $E_t(X_t, A)_i + c$ is non zero with probability $X_{ti}$, while for $X_{ti} < \eta^2$, $E_t(X_t, A)_i + c$ is either non positive and bounded by $-\frac{1}{2}$, or it is positive with probability lower or equal to $X_{ti}$. Eq. (13) uses the condition $X_{ti} \leq \eta$ in the second sum and the upper bound $\eta \leq 1/2$. □

*Proof of Theorem 11.* Combine Lemma 10 with Theorem 2, Corollary 6, and Remark 7. □

# E Proof of Theorem 14

We make use of the following lemma.

**Lemma 18** (Alon et al. 3). *Let $p \in \Delta([k])$. Then*

$$\sum_{i=1}^{k} \frac{p_i}{\sum_{j \in \mathcal{N}(i)} p_j} \leq 4\mathcal{G}_{ind} \log\left(\frac{4k}{\mathcal{G}_{ind} \min_i p_i}\right) .$$

*Proof of Theorem 14.* Starting from Corollary 6 we need to bound the diameter and stability.

$$\text{diam}_F(\mathcal{X}) \leq \frac{k^{1-\alpha}}{\alpha(1-\alpha)} = \frac{k^{\frac{1}{\log(k)}} \log(k)}{1 - \frac{1}{\log(k)}} = \frac{e \log(k)}{1 - \frac{1}{\log(k)}} \leq 2e \log(k) ,$$

where in the last inequality we used the assumption that $k \geq 8 > e^2$. Moving to the stability term. As a reminder we have

$$E_t(X_t, A_t)_i = \frac{\ell_{ti}\mathbb{1}(A_t \in \mathcal{N}(i))}{\sum_{b \in \mathcal{N}(i)} X_{tb}} \text{ for } i \in I_t \text{ and } E_t(X_t, A_t)_i = \frac{(\ell_{ti} - 1)\mathbb{1}(A_t \neq i)}{1 - X_{ti}} + 1 \text{ otherwise}$$

where $I_t = \{i \in [k] : i \notin \mathcal{N}(i) \text{ and } X_{ti} > 1/2\}$. The set $I_t$ is either empty or contains exactly one element, since the action set it the probability simplex. As a slight abuse of notation, $I_t$ denotes either the (possible empty) set or the unique element within. We use Lemma 17 with

$$c = \mathbb{1}(I_t \neq \emptyset)\frac{(1 - \ell_{tI_t})\mathbb{1}(a \in \mathcal{N}(I_t))}{1 - X_{tI_t}} \geq 0 .$$

The Hessian of $F$ is $\nabla F^2(x) = \text{diag}(x^{\alpha-2})$. The non-negativity of $E_t(X_t, A_t) + c\mathbb{1}_k$ ensures that $g_t(X_t, A_t)_i \leq X_{ti}$ almost surely and hence by the definition of the potential $\nabla^{-2}F(z) \preceq \nabla^{-2}F(X_t)$ for all $z \in [X_t, g_t(X_t, A_t)]$,

$$\mathbb{E}_{A \sim P_{X_t}}\left[\sup_{z \in [X_t, g_{tc}(X_t, A)]} \|E_t(X_t, A) + c\mathbb{1}_k\|^2_{\nabla^{-2}F(z)}\right]$$

$$= \mathbb{E}_{A \sim P_{X_t}}\left[\|E_t(X_t, A) + c\mathbb{1}_k\|^2_{\nabla^{-2}F(X_t)}\right]$$

$$= \sum_{i \notin I_t} \mathbb{E}_{A \sim P_{X_t}}\left[(E_t(X_t, A)_i + c)^2 \nabla^{-2}F(X_t)_{ii}\right] + \mathbb{1}(I_t \neq \emptyset)\mathbb{E}_{A \sim P_{X_t}}[\nabla^{-2}F(X_t)_{I_t I_t}]$$

$$\leq 2\sum_{i \notin I_t} \mathbb{E}_{A \sim P_{X_t}}\left[E_t(X_t, A)_i^2 X_{ti}^{2-\alpha}\right] + 2\mathbb{E}_{A \sim P_{X_t}}[c^2]\sum_{i \notin I_t} X_{ti}^{2-\alpha} + 1 .$$

We first bound the $c$ term

$$2\mathbb{E}_{A \sim P_{X_t}}[c^2]\sum_{i \notin I_t} X_{ti}^{2-\alpha} = 2\mathbb{1}(I_t \neq \emptyset)\sum_{i \notin I_t} X_{ti}\left(\frac{1 - \ell_{tI_t}}{\sum_{i \notin I_t} X_{ti}}\right)^2 \sum_{i \notin I_t} X_{ti}^{2-\alpha} \leq 2 .$$

Then we bound the contribution of arms $i$ with $i \notin \mathcal{N}(i)$ and $i \notin I_t$, which implies $X_{ti} \leq 1/2$

$$2\mathbb{E}_{A \sim P_x}\left[\sum_{i:i \notin \mathcal{N}(i) \cup I_t} E_t(X_t, A)_i^2 X_{ti}^{2-\alpha}\right] = 2\sum_{i:i \notin \mathcal{N}(i) \cup I_t} \frac{\ell_{ti}^2 X_{ti}^{2-\alpha}}{1 - X_{ti}} \leq 4 .$$

Finally we bound the remaining term

$$2\mathbb{E}_{A \sim P_x}\left[\sum_{i:i \in \mathcal{N}(i)} E_t(X_t, A)_i^2 X_{ti}^{2-\alpha}\right] \leq 2\sum_{i:i \in \mathcal{N}(i)} \frac{\ell_{ti}^2 X_{ti}^{2-\alpha}}{\sum_{j \in \mathcal{N}(i)} X_{tj}} \leq 2\max_{a \in \Delta([k])}\sum_{i=1}^{k} \frac{a_i^{2-\alpha}}{\sum_{j \in \mathcal{N}(i)} a_j} .$$

We bound the max using Lemma 18:

$$\max_{a \in \Delta([k])} \sum_{i=1}^{k} \frac{a_i^{2-\alpha}}{\sum_{j \in \mathcal{N}(i)} a_j} = \max_{a \in \Delta([k])} \sum_{i:a_i > \exp(-\log(k)^2)} \frac{a_{ti}^{2-\alpha}}{\sum_{j \in \mathcal{N}(i)} a_j} + \sum_{i:a_i \leq \exp(-\log(k)^2)} \frac{a_i^{2-\alpha}}{\sum_{j \in \mathcal{N}(i)} a_j}$$

$$\leq 4\mathcal{G}_{ind} \log\left(\frac{4k \exp(\log(k)^2)}{\mathcal{G}_{ind}}\right) + k \exp(-\log(k)^{-1} \log(k)^2)$$

$$= 4\mathcal{G}_{ind}\left(\log\left(\frac{4k}{\mathcal{G}_{ind}}\right) + \log(k)^2\right) + 1,$$

where in the final inequality we used Lemma 18 on the sub-graph $\{a : X_{ta} > \exp(-\log(k)^2)$ and noted the fact the independence number of a sub-graph of $\mathcal{G}$ cannot be larger than the independence number of $\mathcal{G}$. Combining everything, we have shown that

$$\text{stab}(\mathscr{A}) \leq 8\mathcal{G}_{ind}\left(\log\left(\frac{4k}{\mathcal{G}_{ind}}\right) + \log(k)^2\right) + 9.$$

The proof is completed by tuning the learning rate according to Corollary 6. $\qquad\square$

## F  Proof of Theorem 15

Remember that the potential is $F(x) = \sum_{i=1}^{d} h(x_i)$ where

$$h(x) = \begin{cases} \frac{d}{2}x^2 & \text{if } |x| \leq d^{\frac{1}{p-2}} \\ \frac{p-2}{p-1} d^{\frac{p-1}{p-2}} |x| + \frac{|x|^p}{p(p-1)} + \frac{2-p}{2p} d^{\frac{p}{p-2}} & \text{otherwise}. \end{cases}$$

Before the proof we provide some intuition for this choice of the potential. By the problem setting for $q = \frac{p}{1-p}$, it holds that $\|\ell_t\|_q, \|X_t\|_p \leq 1$. Assuming we have a 'separable' potential $F(x) = \sum_{i=1}^{d} \tilde{h}(x_i)$, we can write the stability term as

$$\|\ell_t\|_{\nabla^{-2}F(z)}^2 = \langle \ell_t \circ \ell_t, (\tilde{h}''(z_i)^{-1})_{i=1,\ldots,d}\rangle \leq \|\ell_t \circ \ell_t\|_{q'} \|(\tilde{h}''(z_i)^{-1})_{i=1,\ldots,d}\|_{p'}.$$

Choosing $q' = \frac{q}{2}, p' = \frac{q'}{q'-1} = \frac{p}{2-p}$, the first factor is bounded by 1 and setting $\tilde{h}''(z_i) = |z_i|^{p-2}$ ensures the second factor is bounded by 1. Unfortunately, this leads to the potential $\tilde{h}(x) = \frac{1}{p(p-1)}|x|^p$, whose diameter can be arbitrarily large. To prevent the potential from exploding, we clip $h''(x)$ at $d$, as shown in Fig. 2. Any upper bound on the second derivative will serve the purpose of decreasing the diameter, however the threshold must be chosen such that the stability doesn't suffer too much. The value $d$ happens to be the lowest value that keeps the stability dimension independent.

Figure 2: $p = 1$: $\tilde{h}''(x)$ and $\tilde{h}(1) - \tilde{h}(x)$ for $p = 1$. Red lines indicate $h''$ and $h$ respectively.

*Proof of Theorem 15.* By the definition of the loss estimator $\hat{\ell}_t = \ell_t$. As usual, our plan is to bound the stability and diameter and then apply Corollary 6.

**Bounding the stability** By definition $h''(x) = \min\{|x|^{p-2}, d\}$. Then by Lemma 3 and the assumption that $E_t(x, a) = \ell_t$ for all $x$ and $a$,

$$\mathrm{stab}_t(x; \eta) \leq \max_{z \in \mathcal{X}} \|\ell_t\|_{\nabla F^{-2}(z)}^2$$

$$\leq \max_{z \in \mathcal{X}} \left( \sum_{i:|z_i| \geq d^{\frac{1}{p-2}}} \ell_{ti}^2 |z_i|^{2-p} + \sum_{i:|z_i| < d^{\frac{1}{p-2}}} \frac{1}{d} \right)$$

$$\leq \max_{z \in \mathcal{X}} \left( \sum_{i=1}^{d} \ell_{ti}^2 |z_i|^{2-p} + 1 \right)$$

$$\leq \max_{z \in \mathcal{X}} \left( \left( \sum_{i=1}^{d} (\ell_{ti}^2)^{\frac{p}{2p-2}} \right)^{\frac{2p-2}{p}} \left( \sum_{i=1}^{d} (|z_i|^{2-p})^{\frac{p}{2-p}} \right)^{\frac{2-p}{p}} + 1 \right) \tag{14}$$

$$= \left( \max_{z \in \mathcal{X}} \|\ell_t\|_q^2 \|z\|_p^{2-p} + 1 \right) \leq 2 \,,$$

where Eq. (14) follows from Cauchy-Schwarz.

**Bounding the diameter** First notice that $F(x) \geq 0$ for all $x \in \mathcal{X}$ and $F(0) = 0$. Hence

$$\mathrm{diam}_F(\mathcal{X}) = \max_{x \in \mathcal{X}} F(x) \,.$$

For arbitrary $x \in \mathcal{X}$ define $J = \{i \in [d] | x_i \geq d^{\frac{1}{p-2}}\}$, $I = [d] \setminus J$ and for any $S \subset [d]$ define the vector $x_S$ as the $|S|$-dimensional vector consisting of entries $(x_i)_{i \in S}$. Then it holds

$$F(x) = \frac{d}{2} \|x_I\|_2^2 - \frac{2-p}{p-1} d^{\frac{p-1}{p-2}} \|x_J\|_1 + \frac{\|x_J\|_p^p}{p(p-1)} + \frac{2-p}{2p} d^{\frac{p}{p-2}} |J|.$$

Maximizing this expression over $x_J$ under the constraints of keeping both the set $J$ and $\|x_J\|_p$ constant is setting all but 1 coordinate in $x_J$ to $d^{\frac{1}{p-2}}$ and shifting all other weight towards a single entry. This follows directly from the fact that $\|x\|_p$ is convex, so the minimum of $\|x\|_1$ under constant $\|x\|_p$ is on the boundary. The optimal $y \in \arg\max_{x \in \mathcal{X}} F(x)$ can therefore only have a single coordinate $i$ such that $|y_i| > d^{\frac{1}{p-2}}$, which we assume without loss of generality is $i = 1$.

$$F(y) = h(y_1) + \frac{d}{2} \sum_{i=2}^{d} y_i^2 \leq h(y_1) + \frac{d^2}{2} d^{\frac{2}{p-2}} \leq h(1) + \frac{1}{2} \,.$$

It follows that

$$\mathrm{diam}_F(\mathcal{X}) \leq h(1) + \frac{1}{2} = \frac{p-2}{p-1} d^{\frac{p-1}{p-2}} + \frac{1}{p(p-1)} + \frac{2-p}{2p} d^{\frac{p}{p-2}} + \frac{1}{2}$$

$$= \frac{1 - d^{\frac{p-1}{p-2}}}{p-1} + d^{\frac{p-1}{p-2}} - \frac{1}{p} + \frac{2-p}{2p} d^{\frac{p}{p-2}} + \frac{1}{2} \leq \frac{1 - d^{\frac{p-1}{p-2}}}{p-1} + 1.$$

We immediately get the bound $\mathrm{diam}_T(\mathcal{X}) \leq \frac{2}{p-1}$. Let $p \leq \frac{3}{2}$, we substitute $z = \frac{p-1}{2-p}$ and get

$$\mathrm{diam}_F(\mathcal{X}) \leq \frac{1 - d^{-z}}{(2-p)z} + 1 \leq 2 \frac{1 - d^{-z}}{z} + 1 \leq 2 \log(d) + 1,$$

where we use the fact that for $z \geq 0$ the term $\frac{1 - d^{-z}}{z}$ is monotonically decreasing in $z$ with limit $\log(d)$ for $z \to 0$.

We have shown that $\mathrm{diam}_F(\mathcal{X}) \leq \mathcal{O}(\min\{\frac{1}{p-1}, \log(d)\})$ and $\mathrm{stab}(\mathscr{A}) \leq \mathcal{O}(1)$. The proof is completed by tuning the learning rate according to Corollary 6. $\square$