[Reviews · NeurIPS 2019]

Reviewer 1



EDIT after rebuttal: Thanks for responding to our questions. I would say the paper is primarily a conceptual one. I view the better regret bounds in the applications as secondary. But application-wise, I agree that it would have greatly helped if they had an application where they give asymptotic improvements better than shaving log factors. But on the flip-side, in one of the applications they give a much simpler proof and replace a log term depending on the horizon with log term depending on the arms which indicates that this connection may have useful applications in the future. Thus, overall I think this paper should be accepted. ----- As described above, the core contribution of this paper is to make a connection between OSMD analysis and the upper-bound on the information ratio. This connection by itself is very interesting. The authors further show that this connection helps in three applications: ( a ) giving a new OSMD based algorithm for the adversarial k-armed bandit which is optimal (upto to lower-order terms) ( b ) bandits with graph feedback and ( c ) online linear optimization where they obtain improved bounds as a result of this connection. Overall, my opinion of this paper is that it is really interesting and technically deep. It answers the open questions in [21] and does so by giving a general connection. Interestingly, now we have optimal algorithms (upto lower-order terms) for the adversarial bandit problem using both Thompson Sampling algorithm (e.g., [21]) and Mirror Descent (this paper)! I had a few minor comments to the author, which I think the paper may benefit from. First, the paper references to [21] multiple times, but I think it would be better to have the relevant theorem statements from that paper in the supplementary, so that the reader doesn't have to go back and forth. For instance, line 168 on page 5. In fact, this is the only theorem that is critical and the full statement might as well be added. Second, I had trouble seeing why line 186-187 in Page 6 holds. The current explanation says plug the said value of \eta into Eq. (4) and then from the definition of stab, it follows. I think you meant it follows from Eq. (5). Moreover, there is another term involving stab(A) and eta. Why does that go away? Nonetheless, I think some more details are in order in these lines. To summarize here are the pros and cons of this paper: Pros: - Makes an important connection between adversarial and bayesian regret. - Establishes optimal OSMD style algorithm for k-armed multi-armed bandit via this connection - Also improves regret bounds for two other problems: bandits with graph feedback and online linear optimization. Cons: - The paper can do a slightly better job of writing and keeping it independent from [21] - Some details were unclear to me, and more explanation was needed.

Reviewer 2



This paper introduces a new connection between Thompson sampling (TS) and online stochastic mirror descent (OSMD), two central algorithms in bandit learning problems. The upshot is that the standard OSMD analysis for linear bandits applies to TS as well. It turns out that a previous analysis due to Russo and Van Roy can be seen as doing this for OSMD with respect to entropy on the simplex, and this work shows that the connection can be extended to any mirror map. This gives a new way to prove regret bounds for Thompson Sampling. I expect this connection to improve and clarify our understanding of linear bandits in the future. A caveat is that these results actually apply only for modified thompson sampling, which requires a prescribed sampling distribution with the same mean as Thompson sampling. In the case of a simplex (ordinary multiarmed bandit) or a full feedback model, there is no difference. However for a general linear bandit problem these can be different, so it remains to be seen how generally the connection applies. As concrete applications, the authors improve the state of the art regret for multiarmed bandit by a constant factor and improve some regret bounds for graphical bandits and full-feedback online learning on Lp balls by small factors. These are fundamental problems, so the improvements appear minor (since the existing results were almost tight already) but bring us closer to the exact optimum rates. The connection looks fruitful for future progress and already clarifies the various performance guarantees obeyed by Thompson sampling for multiarmed bandits (e.g. optimal stochastic and adversarial performance). A small complaint: for the application to online optimization over Lp balls, it seems to me that the stated bound is immediate from prior work in the case of TS. Indeed, in the cases when p is close to 1, we can use the fact that TS regret is bounded by the movement of a martingale in the set. Approximating the Lp metric by L1 and paying the O(1) multiplicative cost for metric distortion when p=1+O(1/log d) seems to show the optimal regret bound for the troublesome case where 1/(p-1) blows up. However, it is still good to have a frequentist algorithm. =========== No updates post-rebuttal.

Reviewer 3



The minimax regret and the Bayesian regret in linear partial monitoring are considered in this paper. While introducing the connection between OSMD policy and Thompson sampling, this paper first proposes a generalization of Thompson sampling, where the expectation of the action (on a vector space) at each round is equal to the expectation of the optimal (fixed) action with respect to the posterior distribution of the loss. An upper bound of the Bayesian regret is provided, which matches the worst-case regret bound of OSMD. The proposed technique is also applied to the analysis of the worst-case regret of OSMD, which improves the known bounds for the classic k-armed bandit problem, a class of bandit problems with graph feedback, and online linear optimization. As far as I understood, the framework (such as the Bayesian setting with non-i.i.d. priors) largely follows that in [21]. There seems to be substantial contribution in the theoretical aspect and the explicit construction of the algorithm (MTS), but I'm not sure about how novel and nontrivial the results of this paper are since I'm not familiar with the settings of this paper and [21]. In fact, the paper has too much weight on the theoretical aspect and I wonder that it may need more emphasis on the practical contribution to make it as a NeurIPS paper rather than a COLT paper. *Remark 5: I do not understand why the MTS matches TS for this case. In MTS, X_{t+1} is determined by the posterior mean of A^*, where not only the posterior distribution but also A^* depends on past observation under the induced filtration. On the other hand, as far as I know, TS depends on the past observation only through the posterior distribution of the loss. Then I want to know why these seemingly different procedures become the same. *Line 80: The description is somewhat misleading around here since it seems from this sentence that linear partial monitoring is just an example of problems considered in this paper while all formal analyses in this paper are given for this problem, although the technique itself seems to be applicable to wider problems. *Equation after Line 133: x seems to be X_t. *Figure 1: The horizontal and vertical axes should be clarified (t and cumulative regret?) ----- Response to the authors' feedback: On the practical contribution, the feedback just refers to the description of the linear partial monitoring that is a generalization of some other problems and there is no additional explanation on how the results (possibly potentially) become benefitial in practice. After re-reading of the paper and the other reviews I became more convinced with the originality of the theoretical contribution of the paper. Still, considering that this paper is currently a purely theoretical one, I also think that the requirement on the theoretical contribution is much higher than other papers with practical contributions and I decided to lower my score. Relation with TS and M-TS: my point does not seem to be caught correctly. For example consider the standard Bayesian setting such that the rewards from each arm is i.i.d. and the prior distribution of the success probability is distributed by, e.g., Beta(1,1), and we have T=11 rounds and have currently observed (5 successes and 0 failure) from arm 1, and (0 success and 5 failures) from arm 2 for the two-armed bandit. Then in the TS that I know, a typical implementation is to generate random numbers theta_1 and theta_2 from Beta(6,1) and Beta(1,6), respectively, and the arm corresponding to the larger theta is chosen at the 11-th round. On the other hand, TS and M-TS that the authors explain, a naive implementation of the algorithm seems to become as follows. - Generate theta_1 and theta_2 from Beta(6,1) and Beta(1,6), respectively. - Generate random numbers (r_{1,1}, r_{1,2},..., r_{1,6}) and (r_{2,1}, r_{2,2},..., r_{2,6}) from Ber(theta_1) and Ber(theta_2), respectively. - If 5+sum_{i=1} r_{1,i} > sum_{i=1}^6 r_{2,i} then pull arm 1, and pull arm 2 otherwise. This algorithm itself seems reasonable since it exploits more when the number of remaining rounds becomes small, but it requires the knowledge of the time horizon and has a weaker connection with the standard TS. Another reviewer informed me that there is a line of works that use this definition of TS, but it is different from TS that most people know and this difference should be clarified. "Equation after Line 133: x seems to be X_t" in my first review: sorry that "Equation above Line 133" is the correct point.

[Author Response · NeurIPS 2019]

We would like to thank all reviewers for their time and feedback.

**Reviewer #1:**   We will add the theorem statement from [21] to the appendix. You are right, this should be Eq. (5).
From Eq. (5) we bound

$$\mathrm{stab}_t(X_t, \eta) \leq \mathrm{ess\,sup}\left(\mathrm{stab}(\mathscr{A})\right)$$

and plug in the learning rate according to Eq. (8).

**Reviewer #2:**   We would like to thank the reviewer again for their detailed comments and observations.

**Reviewer #7:**

- In TS, the posterior distribution of the losses is used to compute the posterior of $A^*$ from which the algorithm samples. In MTS, there is an extra step where we calculate the mean of the posterior and then potentially use a different sampling rule with the same mean. This remark basically says that if the selected sampling rule is actually the posterior of $A^*$, then we can skip the calculation of the mean and the algorithm reduces to regular TS.

- We will clarify Line 80.

- The functions $g_t$ and $f_t$ after Line 133 are intentionally defined for any $x \in \mathcal{X}$ and not only $X_t$, this is necessary to properly define the stability coefficients.

- You are correct, we will add the clarification for Figure 1.

[Meta-Review · NeurIPS 2019]

The paper makes an interesting and seemingly deep connection between a well-known algorithm (OSMD) and "information ratio", a notion that arises in the analysis of some online learning algorithms. This connection can potentially be used as a tool/technique in the analysis of online learning algorithms. The paper presents three applications, so we know that the new tool can actually be used. However, these applications obtain only very marginal improvements over the prior work. The paper is all right as is, but it would be much more impressive if the applications obtained substantial improvements and/or simplifications. There was a substantial discussion among the reviewers and the AC.